# Treatment of Acquired Deforming Hypertonia with Botulinum Toxin in Older Population: A Retrospective Study

**DOI:** 10.3390/toxins16080365

**Published:** 2024-08-16

**Authors:** Pablo Maldonado, Hugo Bessaguet, Cédric Chol, Pascal Giraux, Ludovic Lafaie, Ahmed Adham, Romain David, Thomas Celarier, Etienne Ojardias

**Affiliations:** 1Department of Medical Gerontology, Gier Hospital, 19 rue Laurent Charles, 42400 Saint-Chamond, France; pablo.maldonado@hopitaldugier.fr; 2Department of Physical Medicine and Rehabilitation, University Hospital of Saint Etienne (CHUSE), Bellevue Hospital, 25 Boulevard Pasteur, 42100 Saint-Etienne, France; hugo.bessaguet@univ-st-etienne.fr (H.B.); pascal.giraux@chu-st-etienne.fr (P.G.);; 3Inter-University Laboratory of Human Movement Biology, “Physical Ability and Fatigue in Health and Disease” Team (F-42023), Saint-Etienne “Jean Monnet”, “Lyon 1”, “Savoie Mont-Blanc” Universities, 42000 Saint-Etienne, France; 4Department of Medical Gerontology University Hospital of Saint Etienne (CHUSE), Charité Hospital, 44 Rue Pointe Cadet, 42100 Saint-Etienne, France; cedric.chol@chu-st-etienne.fr (C.C.); ludovic.lafaie@chu-st-etienne.fr (L.L.); thomas.celarier@chu-st-etienne.fr (T.C.); 5Lyon Neuroscience Research Center, Trajectoires Team (Inserm UMR-S 1028, CNRS UMR 5292, Lyon 1 and Saint-Etienne Universities), 42270 Saint-Etienne, France; 6Physical and Rehabilitation Medicine Unit, Poitiers University Hospital, University of Poitiers, 86021 Poitiers, France; romain.david@chu-poitiers.fr; 7PRISMATICS Laboratory (Predictive Research in Spine/Neuromodulation Management and Thoracic Innovation/Cardiac Surgery), Poitiers University Hospital, 86021 Poitiers, France

**Keywords:** acquired deforming hypertonia, contracture, botulinum toxins, older people

## Abstract

Acquired deforming hypertonia (ADH) affects the daily care of numerous nursing home residents. The aim of this study was to analyze the practice, aims, and effectiveness of botulinum toxin injections (BTxis) in the treatment of older patients with contractures, an indication for which BTxis are still underused. Data were extracted retrospectively from medical records regarding population, contractures, and injections. A prospective analysis was conducted to evaluate treatment goals set by goal attainment scaling (GAS) at T0 and at T1, to evaluate the therapeutic effects. We also recorded the occurrence of side effects, using a telephone questionnaire. This study included 41 patients older than 70 years who had received one or more BTxis for the first time between January 2018 and December 2021. Most of the older people we included lived in an institution (66%), manifested severe dependence, and presented significant morbi-mortality (37% of the patients died in the year after the last injection). The main objectives of these injections were purely comfort, without any functional goals. The GAS scores suggested effectiveness for comfort GAS scores. No complications were recorded. This study highlights the BTxis potential to address the needs of a larger number of older patients with ADH.

## 1. Introduction

Limited mobility is the main cause of dependency among older people and contributes to the appearance of what is described as contracture or acquired deforming hypertonia (ADH) [1,2]. This term refers to any joint deformity leading to a reduction in joint amplitude and an increase in resistance to passive mobilization. This results in a limitation of daily activities or in discomfort [2,3]. ADH is estimated to affect between 22% and 88% of residents in nursing homes (NHs) or long-term care units (LTCUs) depending on definitions [3,4]. It causes discomfort and complicates nursing care [2,3]. It also has an overall adverse impact on the quality of life of older people and may even lead to diminished cognitive performance [5,6].

ADH is multifactorial in origin and includes different forms of hypertonia. The term “mixed hypertonia” is generally used to include spastic hypertonia, most often secondary to stroke, and frequently associated with damage to the extrapyramidal nervous system. This term encompasses all Parkinsonian syndromes affecting older people [2]. It also includes oppositional hypertonia associated with damage to frontal loops in neurodegenerative diseases [7,8,9]. Psycho-behavioral symptoms associated with neurocognitive disorders may also contribute to the onset of contracture by aggravating the limitation of mobility [10,11]. In addition to these neuromuscular mechanisms, restricted mobility progressively leads to structural changes in the joints, as well as to structural changes in muscle tissue [2,3,6,12].

The management of ADH begins with primary prevention through early detection, which is crucial for preventing the onset of ADH and allowing for timely intervention and effective treatment planning. This can be easily accomplished at the bedside by primary care providers in NHs and LTCUs [13]. Prevention is also important and can be attained by physical treatments and general measures, as well as by the management of any condition that may lead to spinal irritation (such as urinary tract infection, skin lesion, fecal impaction, etc.). Even after contractures have set in, and despite the fatalism of caregivers faced with these clinical situations, curative treatments remain possible [3]. Intramuscular botulinum toxin injections (BTxis) are the simplest first-line focal treatment available to combat muscle overactivity or hypertonia, the main components of ADH. Other techniques, such as motor nerve neurotomy using alcohol injections or percutaneous needle tenotomies, are effective alternatives but are currently less readily accessible in clinical practice [14,15,16,17].

BTxis guided by ultrasound and/or electro-stimulation are safe even for older people [18]. They can be performed in the consulting room or directly at the patient’s bedside. This focal treatment facilitates limb mobilization and has an analgesic effect [18,19]. However, its practice is not standardized. There are no specific recommendations for dosing BTX in older people to balance efficacy and minimize side effects. Lower doses are generally recommended for initial injections, particularly in patients with comorbidities, but there is no specific guidance related to age [18].

There is a lack of studies describing the use of BTxis for ADH in the older population, despite its apparent efficacy in treating spastic upper limb conditions in institutionalized adults [18,20]. Our study aimed to describe the practice of BTxis in older people referred to the Clinical Gerontology Department of the University Hospital of Saint-Etienne, France, encompassing both acute and chronic geriatric services. Additionally, we investigated the effects of these injections, their specific value for this population, and the potential related adverse events. To achieve these objectives, we conducted a comprehensive retrospective review of patient records to collect detailed data on BTxi treatments administered to older individuals. We systematically gathered specific data points, including treatment protocols, dosing regimens, clinical outcomes, and any adverse effects experienced. These data were analyzed to evaluate the efficacy and safety of BTxis in this population and to identify any patterns or insights relevant to its use in older patients. These strategies were designed to provide a thorough understanding of the practice and impact of BTxis in older adults and to address the existing gap in the literature regarding its use in this population.

## 2. Results

### 2.1. Population Description

This section presents a descriptive analysis of the population with ADH to better understand the challenges it poses and to inform management strategies.

Of the 41 patients included, 37% were living in LTCU, 29.3% in NH, 17% in residences permitting independent living, and 17% at home. The mean age was 84 years [78.5; 88.5]. The Gerontology Groups Iso-Resource (GIR) assessments and French national scales evaluating loss of autonomy for social compensation (see Section 5) were available for 25 patients (70%). Most of these patients were highly dependent: 16 were classified in category GIR 1 (64%), 8 in GIR 2 (32%), and 1 in GIR 3 (4%). Within the overall population, 90% had a Clinical Frailty Score (CFS) of 7 or 8, with 4 patients (10%) having a score of 6.

Comorbidities were mainly neurological. Neurocognitive disorders were present in 78% of patients, in isolated form in 24%. The other main comorbidity was stroke, in 63% of cases. As regards extrapyramidal nervous system conditions, four patients had Parkinson’s disease and one presented cortico-basal degeneration. Pathologies of the musculoskeletal system that could favor the onset of ADH were identified in three patients: retractions following total hip replacement surgery, advanced coxarthrosis, and complex regional pain syndrome (CRPS) affecting the shoulder. One of these three patients presented major neurocognitive disorders, and the other two had a history of ischemic stroke. Population characteristics are fully detailed in Table 1.

### 2.2. Follow-Up Time and Survival

This section presents survival data of patients treated with BTXi for ADH, illustrated by a Kaplan–Meier curve, to assess their survival and understand the degree of frailty in this population.

More than a third of the patients included (n = 15) died within a year of the first BTxi, after a median follow-up of 9 months [3; 15.5]. For patients still alive at the end of the study, the median follow-up (from the first BTxi to December 2021) was 10 months [2.5; 33.7]. All patients were botulinum toxin-naïve at the time of inclusion, 61% subsequently receiving a single set of BTxi, and 39% more than one BTxi. Follow-up data are shown in Figure 1**.** The survival data presented are not associated with BTxi but reflect the overall survival of patients regardless of this treatment.

### 2.3. ADH Distribution

This section provides a descriptive analysis of the various forms of ADH observed. Over half the patients presented more than one ADH (51%), 10% manifesting joint damage in both an upper and a lower limb. The ADH affected the lower and upper limbs almost equally (in 66% and 63% of patients, respectively), with four patients presenting concomitant lower- and upper-limb disorders. The most common types of ADH were closed-hand and claw fingers (29%), hip adductum (22%), triple lower limb flexion (20%), and shoulder adductum (17%). ADH was associated with cervical dystonia treated with BTxis in three patients. Further details are provided in Table 2.

### 2.4. Botulinum Toxin Use

Here, we provide a descriptive analysis of the botulinum toxin injections administered, along with associated therapeutic and non-therapeutic treatments.

Botulinum toxin was injected into a median of 6 [4; 7] muscles per session. The median dose injected during the first injection varied according to the type of botulinum toxin used: 300 units [250; 350] for OnabotulinumtoxinA (Botox^®^, AbbVie Inc., North Chicago, IL, USA), 310 units [250; 400] for IncobotulinumtoxinA (Xeomin^®^, Mertz Pharmaceuticals GmbH, Frankfurt, Germany), and 1200 Speywood units (single injection) for AbotulinumtoxinA (Dysport^®^, Ipsen Biopharm Ltd., Wrexham, UK).

Anticoagulant therapy was ongoing in 23% of patients and was suspended 24 to 48 h before the BTxi, according to French guidelines [21,22]. Anti-platelet-aggregation therapy was ongoing in 17% of patients and was not interrupted except in the case of one patient.

Regarding the measures taken to improve the tolerance to the BTxi, 37% of patients received a lidocaine cream plaster in addition to ice cooling of the treated area. Hypnosis and sophrology techniques, performed by a qualified nurse, were used for 24% of patients. Only three patients (7%) required inhalation of an equimolar mixture of oxygen and nitrous oxide (KALINOX^®^, Air Liquide Santé International, Paris, France).

### 2.5. Goals and Objectives

In this section, we assess the goals that BTxis aims to achieve using the GAS score, focusing on the key goals of older people with ADH.

The treatment goals set (with the corresponding goal attainment scaling (GAS) criteria, see Section 5) were to alleviate pain (“reduce pain”) for 32 (78%) patients, to reduce the difficulty of daily care (“facilitate hygiene care” or “facilitate dressing”) for 26 (63%) patients, to improve passive mobilization (“make mobilization easier”) for 22 (49%) patients, to improve the skin complications of ADH (“moisture-associated skin damage” or “pressure ulcers” or “heal mycosis”) for 19 (46%) patients, and to facilitate comfortable positioning, sitting up (“make sitting up easier”), for 9 (22%) patients. The goals set and GAS criteria defined are detailed in Table 3.

### 2.6. Treatment Outcomes

This section provides an estimate of treatment efficacy using the GAS scale, both for all patients (combining all objectives) and for each objective category.

Concerning patient assessment of overall treatment efficacy, given that the patients did not all choose the same number of goals, their T-scores before BTxi (at T0) varied, with a median of 22.64 [20.96; 25.20]. After the BTxi (at T1), the median T-score was 44.62 [37.28; 63.68], representing a statistically significant gain of 19.12 ± 11.26 T-score points (*p* < 0.05).

The mean GAS T1 score was significantly improved compared to GAS T0 score concerning the goals of pain reduction (−0.46 ± 1.10, *p* < 0.0001), facilitation of nursing care (−0.6 ± 1.09, *p* < 0.0001), mobilization improvement (−0.30 ± 1.17, *p* < 0.0001), and skin complication improvement (−0.56 ± 0.72, *p* < 0.0001), but it was not significantly improved with respect to the goal of positioning improvement (−1.4 ± 0.89, *p* = 0.5). GAS T1 assessments by goal and the distribution of GAS T1 scores for each goal in the patient population concerned are shown in Figure 2. The results of the T1 GAS assessment are detailed in Table 3.

### 2.7. Questionnaire Survey

This section provides a descriptive summary of the responses to the questionnaires from patients and/or their families regarding the benefits and tolerability of BTxi.

The follow-up questionnaire was completed for 14 of the 20 patients alive at the time of the telephone call, giving a response rate of 70%. In 14% of cases, the respondent was the patient; in 57%, the nurse in charge of the patient’s daily care; in 14%, the attending physician; and in 14%, the designated proxy person. Concerning patient quality of life, 14% of the respondents felt that this had been improved by the injections, 43% that it had been moderately improved, and 43% that it had not been improved at all. Ratings of the overall benefit of the injections resulted in a mean score of 3.8 ± 2.55 on a numerical scale from 0 to 10.

Compliance with the mobilization exercises recommended during the consultation was poor, with 71% of patients being unable to benefit from these. The reason given was lack of staff and time. No adverse effects were reported in connection with the intramuscular injections (e.g., hematoma, redness, itching, pain at the injection site) or with botulinum toxin (e.g., muscle weakness, swallowing difficulty, fatigue). Only one other adverse effect (pruritus of non-dermatological origin) was reported by a treating physician, with no certainty as to its causal link with previous injections. The results of the questionnaire survey are detailed in Table 4.

## 3. Discussion

This study aimed to describe the geriatric population treated with BTxis for the management of ADH, the concomitant disorders identified, the methods used to perform the injections, and also to assess the clinical effect of these injections.

Most of the 41 patients included in our study resided in institutions (66%) and were highly dependent (96% of those with a GIR assessment being assigned to GIR categories 1 or 2, and 90% having a CFS of 7 or 8). The main risk factors associated with loss of autonomy were neurocognitive disorders (78%) and stroke sequelae (63%). Patients come to our geriatric Physical Medicine Rehabilitation department for the management of multiple and diffuse ADH, BTxis being mainly prescribed for reasons of comfort (pain relief, facilitation of daily care, and more satisfactory positioning).

The population included had a high morbidity and mortality rate, with over a third of patients (37%) dying within a year of the first BTxi. Apart from the impact of ADH prevalence within frail institutionalized populations, limited post-injection survival may be attributed to the delayed identification of this population. Lack of awareness regarding this condition and the therapeutic options available, as well as pessimistic beliefs concerning treatment possibilities, and multiple comorbidities contribute to the belated administration of appropriate care [3].

Among the comorbidities presented by patients, stroke was among the most prevalent. Whereas younger patients were more frequently referred to specialized Medical and Rehabilitation units following stroke, thus gaining easier access to the resources (professionals, technical facilities, etc.) needed to perform BTxi; post-stroke referral for patients over 80 years appears to be discriminative, resulting in their transfer to polyvalent rehabilitative care units with reduced access to BTxis treatment [23,24]. However, the results of a dedicated care pathway for these patients are favorable [25,26]. These factors encourage better education of caregivers focusing on this population.

A specificity of our fragile population was its median survival of 9 months after the first BTxi. To avoid the uncomfortable transport of very frail patients, mobile teams of geriatricians and rehabilitation specialists need to be developed. These teams could operate directly in nursing homes, detect ADH at an early stage and administer BTXi at the patient’s bedside. For complex conditions, the use of multimodal therapeutic approaches, such as the combination of BTxi with other focal treatments, such as phenol or alcohol injection and/or percutaneous tenotomy, may be relevant. These three treatment categories are, therefore, complementary and well-tolerated [17,27].

However, to the best of our knowledge, the specific use of BTxi in the elderly population has not yet been extensively reported [20,28]. The use of botulinum toxin at the doses we employed appears to be safe in older people, as only one, non-severe, adverse effect was noted (pruritus of non-dermatological origin), without a confirmed relationship to BTxi. Its good tolerance in the older population has been reported in a study favoring lower doses than those used in younger adults [18,20]. In contrast, our study highlights the tolerance of doses similar to those used in younger populations for the initial injection. This difference in doses is explained by the presence of diffuse ADH affecting more than one limb and often involving large muscles, particularly the proximal muscles of the lower limbs, as already described for muscle spasticity [29]. Our observations are based on previous research conducted by our team, which indicated that similar doses in older patients were safe and effective [18,30]. Although there are no formal guidelines and no specific studies directly comparing doses used in older adults to those used in other populations, the precautions suggested by manufacturers advocate for cautious dosing for the first injection. Nevertheless, under careful monitoring, using doses similar to those employed for younger adults may be justified [28]. Particularly, for older patients who are at the end of life, it is crucial to use adequate doses from the outset to provide the best possible care, considering their reduced life expectancy.

Regarding the mechanism of action of botulinum toxin, the objective is muscle paralysis. Several studies have also highlighted the inherent analgesic effect of BTxis, acting on nociceptive signals present on the neuromuscular spindle, or more directly on neurotransmitters such as substance P and glutamate [31,32,33]. Another hypothesis is that the inhibition of the neuromuscular spindle leads to the relaxation of muscle fascias shortened by the abnormal myofibroblast contractions induced by prolonged immobilization [2]. In addition to its analgesic effect, BTxi block the release of acetylcholine from nerve endings, preventing the activation of the sweat glands and reducing sweat production [19]. Botulinum toxin also modulates neuro-immune skin activity, which provides an anti-inflammatory effect and modulates immune responses in the skin [34]. These effects, including the reduction in sweat production, contribute to the healing of skin lesions associated with moisture and those present in certain ADH, such as closed hands [19].

The present study aligns with other, albeit rare, studies on the same topic. Previous research on BTxis as a treatment for spasticity or paratonia in the geriatric population has demonstrated its safety and supported its efficacy in this age group [18,20,25,28]. The present study, which more broadly evaluates ADH, assesses the efficacy of BTxi for this indication using a personalized objective scale (GAS). In their randomized control trial comparing the efficacy of BTxi to treat spasticity in patients (average age of 68.5 years) living in LTCUs with comorbidities, Lam et al. also reported better improvement in GAS scores, but they did not specify the objectives. The improvement in specific GAS objectives, such as care or mobilization, reported in this study, along with improvements in caregiver burden scales or PROM, supports the effectiveness of BTxi in improving daily care for such frail individuals [20,28]. The efficacy of BTxi may be influenced by the patient’s individual pathogenic context, and may vary according to the treatment indication (targeting predominantly the muscular hypertonic component or retractile tendon damage). The development of a model to study response factors to BTxis would, therefore, be of interest to further personalize the objectives likely to be associated with benefit from BTxi.

Our study has several limitations that should be acknowledged. First, the relatively small number of subjects enrolled limits the generalizability of our findings. The study’s sample size, while reflective of the specific population treated at our institution, may not fully capture the broader variability seen in older adults receiving BTxis for the management of ADH. Additionally, the retrospective design of our study may introduce biases related to data collection and reporting, as we relied on existing medical records, which may not consistently document all relevant clinical details. Another limitation is the absence of a direct comparison between the doses used in our older population and those typically employed in younger adults, which could provide a more robust basis for dose standardization across age groups. Large prospective studies are needed to standardize the use of BTxi in adults, especially in the geriatric population. These studies should compare dosing in different age groups to help establish more precise guidelines. The development of standardized protocols tailored to the specific needs of older adults would also improve the safety and effectiveness of this treatment.

## 4. Conclusions

The results of this retrospective cohort study demonstrate the feasibility and benefits of BTxis in older patients with advanced dependency and with at least one ADH. While efficacy needs confirmation through more rigorously designed studies, this treatment was well-tolerated despite the significant morbidity and mortality of these patients. Our study highlights the need for reinforcing knowledge of this condition and the potential of BTxis to address the needs of a larger number of older patients at an earlier stage.

## 5. Materials and Methods

### 5.1. Population and Data Collection

This was a single-center retrospective study. It included 41 patients over 70 years of age who received one or more BTxis in the clinical gerontology department of a University Hospital between January 2018 and December 2021. The study received approval from the ethics committee of the University Hospital (IRBN 1682021/CHUSTE).

Patient socio-demographic and clinical information, and details of the BTxis performed, were collected from medical records. For each patient, the date of death, if applicable, was extracted from the medical record or otherwise checked using the MatchID tool [35].

### 5.2. Autonomy Assessment

Patient level of autonomy was assessed using the Autonomy Gerontology Iso Resource Group (AGGIR) scale whenever possible (Appendix A). The AGGIR scale is a French national scale evaluating loss of autonomy according to six levels ranging from GIR 1 (bedridden older persons with severely impaired mental functions, for whom the continuous presence of care-givers is essential) to GIR 6 (persons autonomous as regards all everyday activities, occasionally needing help with domestic tasks). It is the validated scale used in France to determine the allocation of social and financial assistance to dependent people [36]. Autonomy was also assessed using the Clinical Frailty Scale (CFS) [37]. This scale ranges from 1 (very fit) to 8 (severely frail), with a final score of 9 denoting terminal illness. The more precise AGGIR score was preferred to describe functional autonomy but was not available for all patients. In contrast, the CFS was calculated for the entire patient population based on available data.

### 5.3. Data Collection on ADH and Injections

As part of the etiological search for ADH, we recorded the possible existence of the following comorbidities: stroke, neurocognitive disorders (Mini-Mental State score < 25/30 or as noted in the patient’s medical record), damage to the extra-pyramidal system (as manifested by Parkinson’s disease or other extrapyramidal disorders), or the presence of a pathology impacting the peripheral effector (e.g., osteoarthritis, fracture). The results concerning all these parameters were expressed as the calculated medians, quartiles, and ranges, or means and standard deviations when applicable.

### 5.4. Prospective Survey

An additional telephone questionnaire was administered by the principal investigators to patients still alive at the end of the study, or otherwise to the family, the main professional caregiver, or the attending physician. The questionnaire, taking 5 min to complete, consisted of 11 questions concerning the impact of the injections on the patient’s quality of life and any modifications of care (Appendix A). The patient’s subjective perception of the overall benefit of the injections and compliance with the recommended stretching exercises, as well as the occurrence of any adverse effects were also recorded. For each question, the mean response for each item was calculated with its standard deviation.

### 5.5. Evaluation of Treatment Outcomes

The goals targeted by BTxi were established as part of a care contract between the patient and the caregiver and the team caring for the patient, using GAS. This scaling system is widely used in rehabilitation, psychiatry, and geriatrics for setting individualized goals and quantifying their successful achievement [38,39].

A pre-injection GAS T0 score of −2 was established for each goal (initial state), signifying that the patients were initially in as bad a condition as they could be in the goal areas assessed. Achievement of the goals set was appraised according to the information provided in the patient’s medical record or by calling the patient or caregiver. The post-injection GAS score (GAS T1) was evaluated for each goal, to assess whether this had been achieved partially (GAS score −1), as expected (GAS score 0), better than expected (GAS score +1), or maximally (GAS score +2). For each GAS objective listed, the score obtained by each patient was determined, as was the mean GAS T1 score for all patients combined. The GAS T1 was collected prospectively during a dedicated consultation.

### 5.6. Statistical Analyses

The global GAS scores for each patient were compared between T0 and T1 by calculating the T-score according to the formula of Kiresuk and Sherman [40,41]. Normal distribution was verified by the Agostino and Pearson test, adapted to variables with numerous ties. The difference between the T-score at T0 and T1 was evaluated by the paired Student’s *t*-test. A difference in T-score > 10 points is considered clinically relevant [41]. The mean gain in gross overall GAS score was also calculated for each patient. Finally, for each GAS objective, the mean GAS T1 score of all the patients combined was compared with the GAS T0 (−2) using a non-parametric Wilcoxon test. Differences were considered significant at *p* < 0.05. The software used for the statistical analyses was GraphPad Prism 9^®^ (GraphPad Software 2023, 225 Franklin Street. Fl. 26. Boston, MA, USA).

Data concerning population characteristics, autonomy assessment, characteristics of ADH and BTxi prospective survey results, and efficacy assessment were expressed as means and standard deviations or medians and minimum and maximum values as necessary.

## Figures and Tables

**Figure 1 toxins-16-00365-f001:**
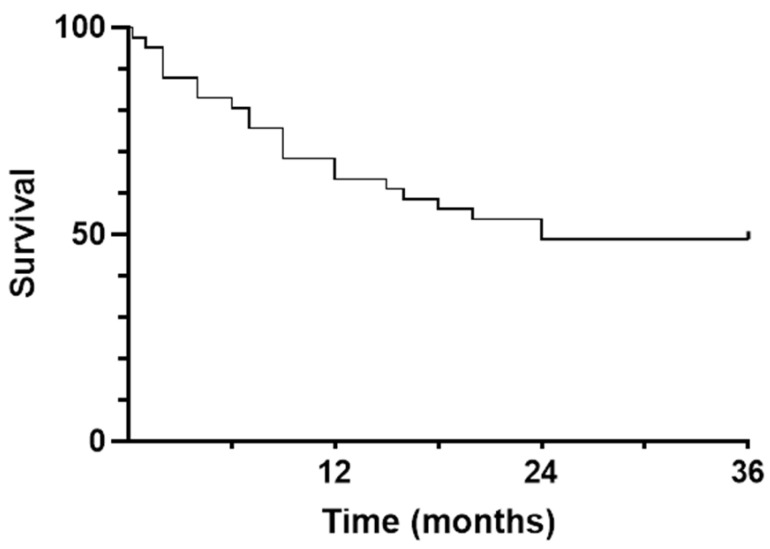
Survival curve of patients receiving botulinum toxin injections (BTxis) to treat acquired deforming hypertonia (ADH); (time in months elapsed between the first injection and death). Legend F1: This survival curve represents the survival of the population during the treatment follow-up period. Time 0 on the *x*-axis corresponds to the first BTxi for each patient (100% of the population alive, *y*-axis). At the end of the study period (24 months), 50% of the population is alive.

**Figure 2 toxins-16-00365-f002:**
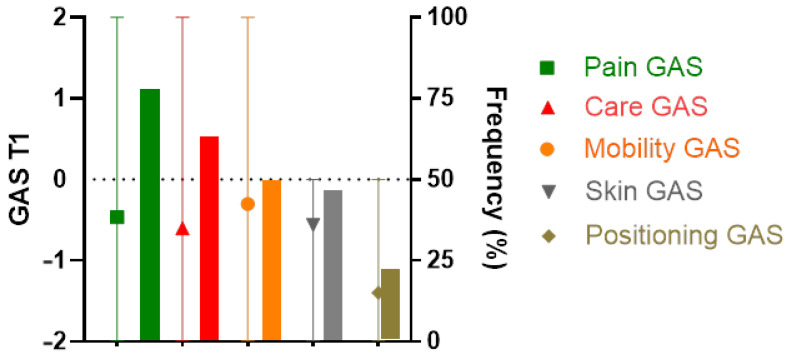
Mean GAS score for each goal (with minimum and maximum values) at the time of individual re-evaluation (T1) (aggregate data for all individuals; symbols) and percentage frequency of each goal among all the goals chosen (histograms). Legend F2: This graph shows the frequency of goals set at the start of treatment with BTxi (right *y*-axis) and the mean of GAS T1 scores by goal category after treatment (left *y*-axis) with minimum and maximum values.

**Table 1 toxins-16-00365-t001:** Population Characteristics.

** *Age:* **	** *Median* **	** *Min; Max* **
	**84**	**71; 100**
	** *n* **	** *%* **
** * Place of residence: * **		
Community-dwelling	7	17
Independent living residence	7	17
Nursing home	12	29
Long-term care unit	15	37
** * Gender: * **		
Male	14	34
Female	27	66
** * Degree of autonomy: * **		
CFS ^a^	41	100
CFS 7 or 8	37	90
CFS 6	4	10
GIR ^b^	25	70
GIR 1	16	64
GIR 2	8	32
GIR 3	1	4
** * Comorbidities: * **		
Stroke sequelae	26	63
Parkinsonian syndrome	5	12
Musculoskeletal disease	3	7
Cognitive impairment	32	78
Associated cognitive impairment	21	51
Isolated cognitive impairment	11	28

^a^ CFS: Clinical Frailty Score (7: severely frail; 8: very severely frail). ^b^ GIR: Gerontology Groups Iso-Resource level (GIR 1: confined to bed, mental functions severely impaired, continuous care essential; GIR 2: either confined to bed or armchair, mental functions not fully impaired, but help required for most everyday activities or mental functions impaired, but mobility conserved; GIR 3: mental autonomy fully or partially conserved, mobility partially conserved, but help required several times a day for personal hygiene autonomy). Legend T1: Table listing the results of the file review concerning the characteristics of the population in terms of age, place of residence, autonomy, and co-morbidities.

**Table 2 toxins-16-00365-t002:** Locations and types of acquired deforming hypertonia (ADH).

	**n**	**%**
**Lower limb**	**27**	**66**
Hip adductum	9	22
Hip flexum	3	7
Equinovarus	3	7
Knee flexum	2	5
Claw toes	2	5
Triple lower flexion (Hip + knee + ankle flexion deformity)	8	20
**Upper limb**	**26**	**63**
Shoulder adductum	7	17
Elbow flexum	3	7
Closed hand and claw fingers	12	29
Triple upper limb flexion (Arm + wrist + finger flexion deformity)	4	10
**Both lower and upper limb**	**4**	**10**
Associated cervical dystonia	3	7
**More than one ADH**	**21**	**51**

Legend T2: Table summarizing the types and locations of ADH identified. Note that some patients presented with more than one type of ADH.

**Table 3 toxins-16-00365-t003:** Definition of GAS scales and GAS T1 assessments results for each goal for the entire patient population. * GAS score unavailable at T1.

**T1 GAS Score**	**Make Mobilization Easier** **(n = 22)**	**Reduce Pain** **(n = 32)**	**Facilitate Hygiene Care and/or Dressing** **(n = 26)**	**Improve Skin Complications** **(n = 19)**	**Improve Positioning (n = 9)**
**−2**	Mobilization with major difficulties or impossible 18% (n = 4)	Pain resistant to conventional analgesics 16% (n = 5)	Complete daily care not possible 19% (n = 5)	Presence of pressure ulcers or moisture-associated skin damage or heal mycosis 10% (n = 2)	Bed positioning uncomfortable, sitting in a chair impossible 66% (n = 6)
**−1**	Easier but very limited mobilization 14% (n = 3)	Reduction in spontaneous, persistent pain during mobilization 28% (n = 9)	Complete daily care difficult but possible 19% (n = 5)	Moderate improvement in pressure ulcers or moisture-associated skin damage or healing of mycoses 26% (n = 5)	Bed positioning more comfortable, sitting in an armchair impossible 11% (n = 1)
**0**	Easier mobilization within target joint amplitudes 45% (n = 10)	50% pain reduction 31% (n = 10)	Easier and more complete daily care 31% (n = 8)	Healing of pressure ulcers, moisture-associated skin damage or mycoses 47% (n = 9)	Bed positioning comfortable, sitting in a chair possible 11% (n = 1)
**+1**	Mobilizations easier even beyond target joint amplitudes 4% (n = 1)	Improvement allowing a reduction in analgesic treatments 6% (n = 2)	Complete daily care easier than expected 4% (n = 1)		Prolonged sitting possible in an adapted armchair 0%
**+2**	No discomfort during mobilization 9% (n = 2)	Pain improvement allowing discontinuation of level 2 or 3 analgesics 6% (n = 2)	Normal daily care 4% (n = 1)		Prolonged sitting possible on any seat (chair) 0%
Data unavailable *	10% (n = 2)	13% (n = 4)	23% (n = 6)	16% (n = 3)	11% (n = 1)
**Mean GAS T1 score**	**−0.3**	**−0.46**	**−0.6**	**−056**	**−1.4**
**Significance of difference between GAS TI and GAS T0 scores**	***p* < 0.0001**	***p* < 0.0001**	***p* < 0.0001**	***p* < 0.0001**	***p* = 0.5**

* GAS score unavailable at T1. Legend T3: Table listing the treatment objectives and their frequency of selection (one GAS scale per objective). For each objective, the GAS evaluation after BTXi (GAS T1) is shown, including how often the objective was achieved and the corresponding mean score. For some patients, the GAS T1 was unavailable. The difference in GAS between T0 and T1 was significant for all objectives except the installation objective.

**Table 4 toxins-16-00365-t004:** Questionnaire answers.

	**n**	**%**
**Questionnaire responder:**		
**Total**	**14**	**100**
Patient	2	14
Nurse	8	57
Primary Care Physician	2	14
Designated proxy	2	14
**Goals targeted:**		
Reduce Pain	12	86
Facilitate daily care	6	43
Facilitate mobilization	6	43
Improve skin complications	4	28
Improve positioning	1	7
**Treatment impact:**		
None	4	29
Below expectations	3	21
Standard performance	5	36
Exceeded expectations	2	14
**Improved quality of life:**		
Yes, significantly	2	14
Yes, moderately	6	43
No, not at all	6	43
**Improved duration or difficulty of daily care:**		
Yes, significantly	0	0
Yes, moderately	7	50
No, not at all	8	57
**Perform the mobilization exercises alone:**		
Yes	1	7
No	12	86
I don’t know	1	7
**Perform the mobilization exercises with other:**		
Yes	10	71
No	4	29
**Why mobilization exercises were not performed:**		
Patient refusal	1	7
Lack of time	4	29
**Felt need for repeat injections:**		
Yes	5	36
No	8	57
I don’t know	1	7
**Side effects:**	1 *	7
**Benefit rating scale (0 to 10) ^1^:**	3.78	

* pruritus of non-dermatological origin. ^1^ Mean (± 2.54). Legend T4: This table shows the number of responses in the telephone questionnaire conducted with patients and their caregivers.

## Data Availability

The raw data supporting the conclusions of this article will be made available by the authors on request.

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
