# Peer review of "Treatment of Acquired Deforming Hypertonia with Botulinum Toxin in Older Population: A Retrospective Study"

_toxins, 2024, doi:10.3390/toxins16080365_

Round 1
Reviewer 1 Report
Comments and Suggestions for Authors
Comments to the manuscript entitled "Treatment of acquired deforming hypertonia with botulinum 2 toxin in older population: a retrospective study"
Introduction
1) Line 45: The authors mention that "the management of ADH is based primarily on detection". Could the authors please paraphrase this statement? In its current form, it needs to be clarified.
2) The main document should describe the abbreviations to better understand the readers. Line 46, please define "LTCU".
3) Line 55. The authors mention, "BTxi guided by ultrasound and/or electro-stimulation are safe even for older people [18]. They can be performed in the consulting room or directly at the patient's bedside. This focal treatment facilitates limb mobilization and has an analgesic effect". Could the authors mention whether there is information about the doses applied in older adults.
4) The authors need to reinforce their focus on why they carry out this work. Before the Materials and Methods section, the authors must indicate which strategies were used to achieve their objectives.
Results
1) Could the authors please add a brief introduction in each results section, indicating the purpose of each analysis, experiment, or test? It is to guide the reader to understand the importance of the results.
2) Section "2.2. Follow-up time and survival", I wonder if the authors are associating the patient's death with the injection of the toxin; could they confirm this in the text?
3) In Figure 1, please add a brief description of the figure in the legend, indicating what it represents. Please describe the Figure legends in more detail.
4) Section "2.3. Botulinum 114 toxin use" should be corrected to 2.4
5) Section "2.4. Goals and objectives" should be corrected to 2.5
6) 2.5. Treatment outcomes should be 2.6
7) 2.6. The questionnaire survey should be to 2.7
8) I wonder if there is any relationship between doing mobilization exercises and the impact of the treatment.
9) Please add the description of each table and standardize the format.
Discussion
1) The results of the study need to be analyzed and discussed.
2) Line 242: "In contrast, our study highlights the tolerance of doses similar to those used in younger populations for the initial injection." Could you please expand this sentence, since an analysis of the doses used in the present study and those used in other populations was not carried out.
3) Authors should mention the limitations of their study, for example, the number of subjects enrolled, and make suggestions about the prospects for achieving standardization of the application of the toxin in adults, which is the main objective of this paper.
Reviewer 2 Report
Comments and Suggestions for Authors
1. Lines 256-257: the sentence stating, "BTxi also has a vasodilatory effect and reduces sweating," may lead to a misunderstanding about the mechanism by which BTxi reduces sweating. It is important to clarify that the reduction in sweating is not due to a vasodilatory effect but rather to the inhibition of acetylcholine release. The correct mechanism is that BTxi blocks the release of acetylcholine from nerve endings, thereby preventing the activation of the sweat glands and reducing sweat production.
2. It would also be valuable to include information on the broader effects of botulinum toxin, specifically its role in modulating neuroimmune cutaneous activity. This mechanism can contribute additional benefits, such as anti-inflammatory effects and the modulation of immune responses in the skin, which are relevant for conditions beyond just excessive sweating.
Please see this paper: Popescu MN, Beiu C, Iliescu MG, Mihai MM, Popa LG, Stănescu AMA, Berteanu M. Botulinum Toxin Use for Modulating Neuroimmune Cutaneous Activity in Psoriasis. Medicina (Kaunas). 2022 Jun 16;58(6):813. doi: 10.3390/medicina58060813. PMID: 35744076; PMCID: PMC9228985.
